# Evaluation of Vascular Endothelial Function in Children with Type 1 Diabetes Mellitus

**DOI:** 10.3390/jcm10215065

**Published:** 2021-10-29

**Authors:** Karolina Nocuń-Wasilewska, Danuta Zwolińska, Agnieszka Zubkiewicz-Kucharska, Dorota Polak-Jonkisz

**Affiliations:** 1Department of Pediatric Nephrology, Wroclaw Medical Univeristy, 50-367 Wrocław, Poland; nocun-wasilewska@wp.pl (K.N.-W.); danuta.zwolinska@umed.wroc.pl (D.Z.); 2Department of Pediatric Endocrinology and Diabetology, Wroclaw Medical University, 50-367 Wrocław, Poland; agnieszka.zubkiewicz-kucharska@umed.wroc.pl

**Keywords:** diabetic kidney disease, vascular endothelial markers, children, eGFR

## Abstract

Diabetic kidney disease belongs to the major complications of diabetes mellitus. Here, hyperglycaemia is a key metabolic factor that causes endothelial dysfunction and vascular changes within the renal glomerulus. The aim of the present study was to assess the function of the vascular endothelium in children with type 1 diabetes mellitus (type 1 diabetes) by measuring selected endothelial lesion markers in blood serum. The selected markers of endothelial lesions (sVCAM-1, sICAM-1, sE-SELECTIN, PAI-1, ADMA and RAGE) were assayed by the immunoenzymatic ELISA method. The study involved 66 patients (age: 5–18 years) with type 1 diabetes and 21 healthy controls (age: 5–16 years). In the type 1 diabetes patients, significantly higher concentrations of all of the assayed markers were observed compared to the healthy controls (*p* < 0.001). All of the evaluated markers positively correlated with the disease duration, the age, and BMI of the patients, while only PAI-1 and sE-SELECTIN were characteristic of linear correlations with the estimated glomerular filtration rate (eGFR). It can be concluded that endothelial inflammatory disease occurs in the early stages of type 1 diabetes mellitus in children. The correlations between PAI-1, sE-SELECTIN, and eGFR suggest an advantage of these markers over other markers of endothelial dysfunction as prognostic factors for kidney dysfunction in children with type 1 diabetes.

## 1. Introduction

Diabetes is a significant health problem in children, including children who are very young. Since the year 2006, it diabetes has been considered to be one of the main health conditions with the potential of endangering public health around the world [1]. Chronic hyperglycaemia results in the enhanced production of toxic oxygen derivatives that, together with their impaired elimination by the antioxidative systems of the body, cause a specific status to emerge called “oxidative stress” [2,3,4]. This entails modifications in the structures of proteins, lipids, carbohydrates, or nucleic acids, resulting in degenerative changes to tissues, mainly those in the vascular area. Diabetic nephropathy, one of the most serious complications of diabetes mellitus, as a clinical picture of renal microangiopathy, results from metabolic disturbances (the “glucotoxicity” effect) and coexisting inflammation. Pathomorphologic changes in the renal structure may be present at the time of diabetes diagnosis, resulting in initial hypertrophy and hyperfunction at the early stages and throughout albuminuria, eventually leading to end-stage renal failure. 

An enhanced, non-enzymatic glycation of the basal lamina has been observed from both the glomeruli and the mesangial matrix and shows the formation of advanced glycation products that contribute to the loss of the negative charge of the filtration membrane, increased intraglomerular pressure, and glomerular hyperfiltration [5]. At present, albuminuria is considered the marker of generalised vascular endothelium damage [6]. However, the increased concentrations of inflammation indices can already be found during the early period where changes occur in the kidneys, which occur long before glomerulosclerosis and which may disappear after the metabolic balance of the disease is regained. The risk of the occurrence and progression of chronic complications principally depend on the effective metabolic control of diabetes mellitus; therefore, people with well-controlled disease are much less likely to be endangered by chronic complications. In addition, extraglycaemic risk factors of sclerosis, such as hypertension, hyperlipidemia, and obesity, also influence the progression of vascular changes [7,8,9]. 

The vascular endothelium is the largest endothelial structure in the body and produces numerous substances that maintain vascular homeostasis. Endothelial products, depending on their functions, are divided into three categories. These include vasomotor substances (e.g., asymmetric dimethylarginine-ADMA), substances affecting coagulation and fibrinolysis (e.g., plasminogen activator inhibitor-PAI-1), and substances regulating vascular permeability and inflammatory processes (e.g., E-selectin, vascular adhesion molecule-VCAM-1, intercellular adhesion molecule-ICAM-1). These markers, together with circulating serum receptors for advanced glycation end-products-RAGE, are considered as new biomarkers for endothelial damage and atherosclerosis development. This is because their high concentrations have been observed in cardiovascular diseases and because they have been shown to correlate with the presence of cardiovascular risk factors. What is more, their baseline value decreases after statin therapy [10]. The very fact that their expression is increased by inflammatory mediators such as IL-1, TNF, and LPS is certainly not without significance [11,12].

Taking into account the finding that chronic hyperglycemia is the most significant, unfavourable factor and that it affects the endothelial cells in particular, the identification of reliable markers of progressive endothelium dysfunction seems to be a primary objective. The existing knowledge gap as to whether biomarkers of inflammation and endothelial dysfunction are associated with prognosis in type 1 diabetes needs to be filled. [9].

### The Aim of the Study

The aim of this study was to assess the vascular endothelium function in children with type 1 diabetes mellitus (type 1 diabetes, DM1) based on selected markers of endothelial lesions such as sVCAM-1, sICAM-1, sE-SELECTIN, PAI-1, ADMA, and RAGE. The tests performed in the present research were supplemented with an analysis of the effects of type 1 diabetes duration, the degree of its metabolic balance, as well as of the effects of the patient’s age, sex, and BMI on the extent of endothelial damage.

## 2. Materials and Methods

The prospective observational study involved sixty-six (66) children (36 boys and 30 girls) with diagnosed type 1 diabetes who were being treated at the Department of Pediatric Endocrinology and Diabetology at the University Hospital. The presence of other chronic diseases that could have been either inflammatory or influential for the selected parameters were excluded.

The control group included twenty-one (21) healthy children (10 boys and 11 girls). The control children were hospitalized at the Department of Pediatric Nephrology at the University Hospital due to suspected urinary tract abnormalities or bedwetting. On the basis of the diagnostic examinations performed at that time, the above-mentioned abnormalities were excluded.

Anthropometric measurements were conducted in both the group of children with type 1 diabetes (the study group) and the control group and included body height and weight with body mass index (BMI) estimation measurements as per the BMI centile charts developed by the WHO (see the WHO Child Growth Standards). The study patients were qualified to four (4) groups depending on their nutrition status: UN—undernourished children (BMI < 15 centiles), N-properly nourished children (BMI < 15–85 centiles), OV—children with overweight (BMI 85–90 centiles), and O—obese children (BMI ≥ 90 centiles).

Blood and urine from the patients were used as the research material. Blood was collected in the morning from the veins of the elbow fossa into tubes without anticoagulant (the so-called “clot”) a minimum of 12 h after the patient had last ingested food or fluids. Then, the blood samples were centrifuged for 15 min at +4 °C at 1000× *g*. The obtained serum, which was necessary for the determination of the vascular endothelial damage markers (sVCAM-1, sICAM-1, sE-SELECTIN, ADMA, PAI-1, RAGE), was stored in Eppendorf tubes in the amount of approx. 400 µL and at a temperature −70 °C until the planned determinations were made. 

Albuminuria was assessed by the immunoturbidimetric method, having collected a 24 h urine sample.

All of the studied endothelial inflammation markers were assayed by means of the immunoenzymatic ELISA test and by its variant, i.e., the so-called “sandwich” ELISA (a double-binding test) in particular, where the antigen is bound between two layers of antibodies [13,14]. The assays were conducted using the following sets: R&D Systems (sICAM-1, sVCAM-1, sE-SELECTIN, PAI-1) and Wuhan EIAab Science (RAGE, ADMA), according to the manufacturer’s instructions. The degree of metabolic DM control was evaluated using the HbA1c concentration measured with high-performance liquid chromatography (HPLC). The eGFR value was assessed on the basis of serum creatinine concentration and was assayed by the Jaffe method and the Schwartz formula.

k-factor: 0.33—low birth weight infants; 0.45—normal birth weight infants; 0.55—children 2–12 years; 0.55—girls 13 years and older; 0.70—boys 13 years and older [15,16]. The results were related to age- and sex-specific norms for children.

### Ethical Issues

A statistical analysis was conducted by means of the Statistica software (TIBCO Software Inc. (2017). Statistica (data analysis software system), version 13. http://statistica.io (accessed on 25 August 2021)). A single-factor analysis of variance (ANOVA) was applied for the statistical material analysis using Tukey’s post hoc test or the Mann–Whitney nonparametric test. The correlation among continuous features was also determined using the Spearman correlation coefficient. The studied continuous features were characterised by their distribution parameters, i.e., the mean value, the standard deviation (SD), and the sample size (N). 

Two-sided *p*-values less than 0.05 were considered significant. Calculated p-values were not adjusted for multiple testing. Standard boxplots with bold lines indicating median value were made, with the upper and lower edges of the box showing the first quantile and third quantile results. Black dots are individual data points.

## 3. Results

The clinical characteristics of the study groups are present Table 1. Compared to the control group, the DM1 group had significantly higher age, BMI, CRP, and percentage of males (see Table 1).

The children with type 1 diabetes demonstrated significantly higher concentrations of all of the studied markers of vascular endothelial damage, i.e., sICAM-1, sVCAM-1, sE-SELECTIN, ADMA, PAI-1 and RAGE, when compared to the corresponding values in the control group (see Table 1).

For all of the studied biomarkers, the minimal values in the DM1 group were higher than the maximal values in the control group: the sICAM-1 ranged from 102.55 to 117.8 in control group vs. from 162.41 to 425.35; sVCAM-1 ranged from 313.86 to 357.27 in the control group vs. from 400.20 to 762.65; sE-SELECTIN ranged from 29.12 to 35.38 in the control group vs. from 41.94 to 79.95; ADMA ranged from 68.49 to 75.81 in the control group vs. from 104.71 to 693.67; PAI-1 ranged from 6.02 to 6.82 in the control group vs. from 12.82 to 19.38; and RAGE ranged from 80.10 to 88.12 in the control group vs. from 114.47 to 796.66. Thus, it was possible to predict DM1 based on each biomarker with 100% accuracy.

A positive, linear correlation was observed in the study group between the concentrations of the assayed endothelial damage markers and the disease duration. A significant increase in the concentrations of those markers was found already at very early stages of the disease, i.e., from the moment of its diagnosis, while the levels of C-reactive protein (CRP), the inflammation marker in the body, did not differ statistically significantly in relation to the control group (0.8 ± 1.2 vs. 2.97 ± 4.53 mg/L; *p* = 0.161) (see Figure 1a–g).

Table 2 lists the data based on DM1 duration (Group I, II, III) and the control group (Control).

A positive correlation was also proven between all of the studied endothelial damage markers and the age of the patients; however, statistically significant differences were observed in the youngest children when compared to the patients in the intermediate age group, a finding that specifically concerned ADMA concentrations (196.99 ± 135.66 vs. 331.71 ± 246.17 ng/mL; *p* ≤ 0.01) (see Figure 2).

In addition, a linear correlation was identified between the concentrations of all of the assayed endothelial dysfunction markers and the BMI values of the patients. The children with diabetes mellitus and who were underweight (the “U” subgroup) presented statistically significant concentrations of sICAM (209.94 ± 70.31 vs. 279.44 ± 108.54 ng/mL; *p* ≤ 0.05), sVCAM-1 (481.18 ± 99.39 vs. 568.94 ± 141.01 ng/mL; *p* ≤ 0.05), and RAGE (231.46 ± 198.32 vs. 414.20 ± 298.22 pg/mL; *p* ≤ 0.05) when compared to the children with a normal body weight (the “N” subgroup), while the patients who were overweight (the “O” subgroup) demonstrated much higher albuminuria vs. the normal-weight patients (23.67 ± 16.08 vs. 15.81 ± 14.41 mg/day; *p* ≤ 0.05); however, there was no linear correlation between those parameters (see Figure 3a–d).

Taking into account the degree of metabolic control of diabetes mellitus, the highest concentrations of the studied endothelial damage markers were observed in the children with the lowest HbA1c values, i.e., between 6.5 and 8.9%, while the lowest ones were observed in the children with the worst glycaemic control (HbA1c ≥ 14%) (see Figure 4a–f). In case of sVCAM-1 and PAI-1, there was a negative correlation between those two markers and HbA1c concentration levels.

Urine albumin concentration measurements were available for 55 DM1 patients. The median concentration of urine albumin was 12.9 mg/24 h, with an interquartile range 5.25 mg/24 h–23.85 mg/24 h. In 10 patients (18.18%), the urine albumin concentration exceeded 30 mg/24 h. There were not any statistically significant differences in the levels of the biomarkers between the groups with and without albuminuria.

In turn, while comparing the results of the patients with reference to their sex, it was found that the boys were characterised by statistically significantly higher glomerular filtration rates vs. the studied girls. Otherwise, no statistically significant differences were demonstrated between the boys and the girls regarding the endothelial damage markers.

All of the studied vascular endothelial dysfunction markers correlated with one another (see Figure 5).

Simple linear regression was performed to predict eGFR based on each one of studied biomarkers. Among them, sE-SELECTIN had the highest R2 = 0.09 and had a regression coefficient = −0.75, SE = 0.28, and intercept = 209.28, *p* = 0.0088. In the multiple linear regression analysis, sE-SELECTIN was proven to be an independent predictor (*p* = 0.033) of eGFR. The model was adjusted for age, gender, and duration of DM1.

## 4. Discussion

Although many insights and knowledge about diabetes mellitus have been developed over the last 5000 years, there are many still unanswered questions [17,18]. It is not known what underlies the destruction of pancreatic β-cells. It has, however, been demonstrated that persistent hyperglycaemia leads to progressive vascular endothelial dysfunction which, in turn, underpins the development of diabetic micro- and macroangiopathy [19,20]. Since the early diagnosis of the disease is so important for the prevention of these dangerous complications, the identification of reliable endothelial dysfunction markers should become our priority. In this study, we used the serum from patients to determine the biochemical substances produced by the endothelium, such as vascular cell adhesion molecules (VCAM-1), intercellular adhesion molecules (sICAM-1), selectin E (sE-Selektin), asymmetric dimethylarginine (ADMA), plasminogen activator inhibitor 1 (PAI-1), and receptors for advanced glycation end products (AGEs), because their concentrations increase rapidly in states of cellular stress.

In the analysed material, the children with type 1 diabetes demonstrated significantly higher concentrations of all six studied endothelial damage markers (sICAM-1, sVCAM-1, sE-SELECTIN, PAI-1, ADMA, RAGE) when compared to the healthy controls. A significant increase in the concentrations of the markers was already found in the patients with newly diagnosed diabetes mellitus, i.e., at a very early stage of the disease, while the systemic inflammation index (CRP) was normal. The longer the disease duration was, the more distinctive the increase in the concentrations of the above-mentioned inflammation markers was. This was confirmed by the high positive correlation values between the concentrations of the studied markers and the disease duration. Moreover, the markers grew linearly against one another. This observation confirms earlier reports that diabetes duration is an important risk factor for the development of chronic diabetes complications, which are characterized by chronic subclinical endothelial inflammation [21,22].

Taking into account the degree of metabolic balance in diabetes mellitus, the highest concentrations of the studied endothelial damage markers were observed in the children with the lowest HbA1c values, while the lowest ones were observed in the children with the worst glycaemic control. In addition, in the case of sVCAM-1 and PAI-1, there was a negative correlation between those two markers and HbA1c concentration levels. Since poor glycaemic control is a significant risk factor for complications in diabetes mellitus, an inverse relation could have been expected. However, it should be noted that the diabetic patients with the best glycaemic control were also characterised by the lowest glomerular filtration rates and had been suffering from diabetes mellitus longer than those with the statistically significantly lower concentrations of endothelial damage markers. In fact, there was a strong positive correlation between the studied endothelial damage markers and the duration of diabetes mellitus, while the marker levels decreased linearly with the growing glomerular filtration rate. Observations from other authors regarding the issues have been divided: the same researchers, while evaluating the concentrations of various endothelial inflammation markers, simultaneously obtained positive and negative correlations with glomerular filtration rates [20,23]. Moreover, the correlation between the levels of the markers and diabetes mellitus duration was often not indicated at all, even showing other factors, such as the metabolic control of DM or the patient’s age [24,25]. Moreover, it should be underlined that it is not only hyperglycemia that has a damaging effect on the endothelium; hypoglycemia also induces inflammation [26,27,28]. Therefore, increased risk of chronic complications resulting from endothelial inflammation is a consequence of all glycemic fluctuations (so called glycemic variability) should be considered in diabetes, as these fluctuations undoubtedly have a negative impact on the endothelium [29,30,31]. This may explain the inverse relationship between HbA1c and the concentration of the investigated markers of inflammation. Frequent episodes of hypoglycaemia lower the level of HbA1c, but at the same time, hypoglycaemia also negatively affects endothelial condition. Furthermore, in our cohort, the majority of children with the highest HbA1c level were the newly diagnosed patients. Such disproportion might have biased the result, as in those patients, the levels of studied parameters were the lowest, indicating the inverse correlation of who had had the disease for longer.

In our study, we also examined the impact of the patient’s age on the status of the vascular endothelium. Similar to other studies, that correlation proved to be an important factor for the progress of vascular changes, indicating a positive, linear correlation with all of the studied markers of endothelial damage as well as a positive correlation with the Y variable, which corresponds to the endothelial inflammation intensity [24,32]. Although the concentrations of all of those indices demonstrated the lowest values in the subgroup with the youngest patients, statistically significant differences were only found for ADMA. This result proves the suitability and usability of the asymmetric dimethylarginine concentration assay for the prognosis of early changes in the endothelium, especially in younger patients. Other significant differences that were analysed with regard to the patient’s age concerned disease duration, the metabolic degree to which the patient’s diabetes mellitus was balanced, and albuminuria. This implies that the oldest children would have been suffering from the disease for the longest period of time, and they were characterised by the worst glycaemic control and the highest albuminuria. Such results should certainly not come as a surprise. They once again confirm the mutual correlation of the above-mentioned parameters, as both poor glycaemic control and DM duration are the risk factors of diabetic kidney disease and consequently contribute to enhanced albuminuria [32,33].

In turn, while comparing the results of the patients with reference to their sex, it was only found that the boys had been characterised by statistically significantly higher glomerular filtration rates vs. those of the studied girls. That result was in line with the expectations since eGFR, when taking into account the parameters of the maturing body (the patient’s height, age and sex) as calculated by the Schwartz method, requires higher values [15]. On the other hand, no statistically significant differences were demonstrated between the boys and the girls regarding either the endothelial damage markers or the other evaluated parameters.

In addition, the in-house research results have allowed us to determine that children with diabetes mellitus and who are underweight present statistically significantly lower concentrations of sICAM, sVCAM-1, and RAGE when compared to children with a normal body weight. What is more, a linear correlation has been confirmed between the BMI values of the patients and the concentrations of these and other markers of endothelial dysfunction. Many publications also confirm such correlations [34,35,36,37,38,39]. On the other hand, the children who are overweight demonstrated significantly higher albuminuria vs. normal weight children, who had no linear correlation between the parameters. Moreover, positive correlations with BMI were found, both in terms of the patient’s age and the duration of the patient’s diabetes mellitus. Increased concentrations of endothelial dysfunction markers accompanied by increased BMI values may indicate the inflammatory state of the endothelium in the course of the developing metabolic syndrome (obesity as the risk factor of sclerosis). It could also be the case that the concomitance of these two conditions increases the risk of angiopathic complications, including diabetic kidney disease.

In contrast, while analysing the correlations between those markers and the other parameters, based on the assays in all of the patients (i.e., both in the study group and the control group), our attention was driven by the positive correlations between both PAI-1 and eGFR and between sE-SELECTIN and eGFR. While all of the studied markers revealed positive correlations with the risk factors for vascular complications (BMI, the patient’s age, disease duration), only the two above-mentioned factors demonstrated linear correlations with the kidney damage factor (eGFR), and they thus seem to be more suitable in the prognoses and detection of early unfavourable changes in the kidneys.

Musial and Zwolinska reported that the concentrations of matrix metalloproteinases (MMPs) and their tissue inhibitors (TIMPs) correlate not only with the markers of inflammation, e.g., e selectin, but also with eGFR, thus indicating increased inflammation and endothelial dysfunction in patients with renal failure [40]. Similar findings were reported by Gheissari et al. and Meamar et al. [41,42]. As such, it may be assumed that markers of inflammation, i.e., e-selectin, may act not only as the predictors of cardiovascular complications in chronic kidney disease but also to predict late diabetes complications, including nephropathy.

To sum up, the assayed endothelial dysfunction markers proved the presence of inflammatory condition in the endothelium, which was already at the very early stages of the disease, i.e., from the time of its diagnosis, when the inflammation marker (CRP) was not yet elevated. The enhanced inflammation of the endothelium depended on the already well-known risk factors for vascular complications, namely disease duration, the patient’s age, or his/her BMI. What is more, all of the studied markers demonstrated positive linear correlations between one another, while their increasing concentrations reflected progressive endothelial inflammation. Therefore, the essential issue is whether the evaluated markers are sensitive and specific enough to be used for the assessment of vascular endothelial inflammation and thereby for the estimation of the risk of vascular complications in diabetes mellitus. Such studies should be conducted in children in future to fully determine this. When looking at this issue in the context of nephrological complications, attention should be given to the linear correlations between PAI-1 and eGFR as well as to those between sE-SELECTIN and eGFR, as such results may suggest a certain advantage of PAI-1 and sE-SELECTIN over the other endothelial dysfunction markers, especially regarding the identification of early changes in the kidneys.

## 5. Conclusions

In the patients with type 1 diabetes, statistically significantly higher concentrations were demonstrated for all the assayed markers when compared to the corresponding values in the control group.A significant increase in the concentrations of those markers was already observed at the early stages of the disease.All of the evaluated endothelial dysfunction markers were positively correlated with the disease duration, the age of the patients, and their BMI, while only PAI-1 and sE-SELECTIN were characteristic of linear correlations with the estimated glomerular filtration rate (eGFR).

## Figures and Tables

**Figure 1 jcm-10-05065-f001:**
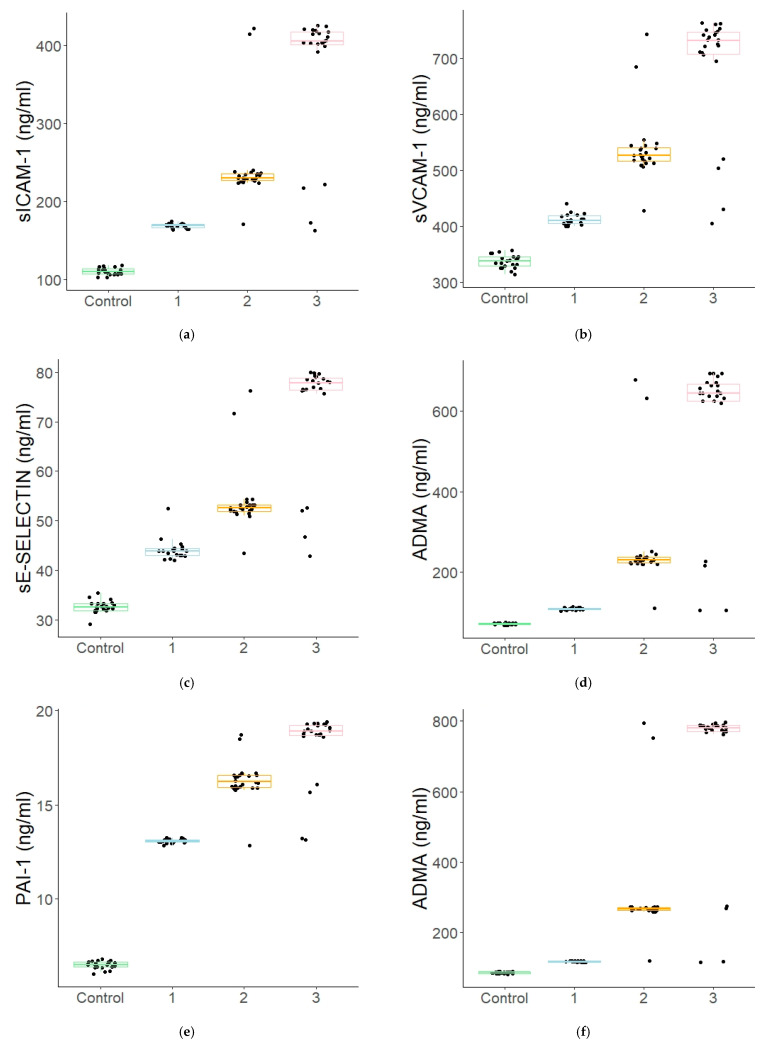
(**a**–**g**). A comparison of the distribution of selected parameters vs. DM1 duration. Boxplot of values for each subgroup. ((**a**)—the mean sICAM-1 values: 110.48 ± 4.53 vs. 168.47 ± 3.05 vs. 243.97 ± 55.18 vs. 372.85 ± 85.31 ng/mL; (**b**)—the mean sVCAM-1 values: 337.33 ± 11.73 vs. 412.56 ± 10.29 vs. 538.29 ± 59.77 vs. 687.26 ± 107.84 ng/mL; (**c**)—the mean sE-SELECTIN values: 5.43 ± 0.21 vs. 44.17 ± 2.27 vs. 54.01 ± 6.5 vs. 72.84 ± 11.56 ng/mL; (**d**)—the mean ADMA values: 71.91 ± 2.31 vs. 109.32 ± 2.38 vs. 261.05 ± 124.20 vs. 569.99 ± 193.44 ng/mL; (**e**)—the mean PAI-1 values: 6.51 ± 0.21 vs. 13.07 ± 0.11 vs. 16.29 ± 1.04 vs. 18.21 ± 1.84 ng/mL; (**f**)—the mean RAGE values: 84.94 ± 2.27 vs. 116.78 ± 1.53 vs. 301.95 ± 148.48 vs. 679.45 ± 230.59 pg/mL; and (**g**)—the mean CRP values: 0.8 ± 1.2 vs. 2.97 ± 4.53 vs. 2.78 ± 4.77 vs. 4.82 ± 7.92 mg/L). Legends: Control—control group; 1—the patients with newly diagnosed diabetes mellitus (DM1 duration < 1 year), *n* = 19; 2—the patients with DM1 duration of 1–5 years, *n* = 24; 3—the patients with DM1 duration of at least 5 years, *n* = 23.

**Figure 2 jcm-10-05065-f002:**
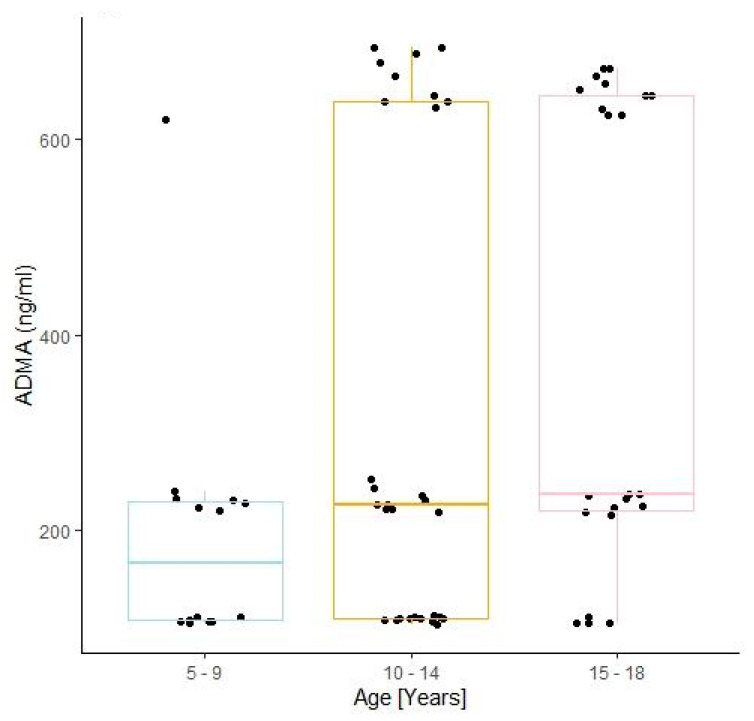
A comparison of the ADMA values depending on the patient’s age (196.99 ± 135.66 vs. 331.71 ± 246.17 vs. 397.40 ± 238.84 ng/mL). Boxplot of values for each subgroup. Legends: patients aged 5–9-years-old, *n* = 14; patients aged 10–14-years-old, *n* = 30; patients aged 15–18-years-old, *n* = 22.

**Figure 3 jcm-10-05065-f003:**
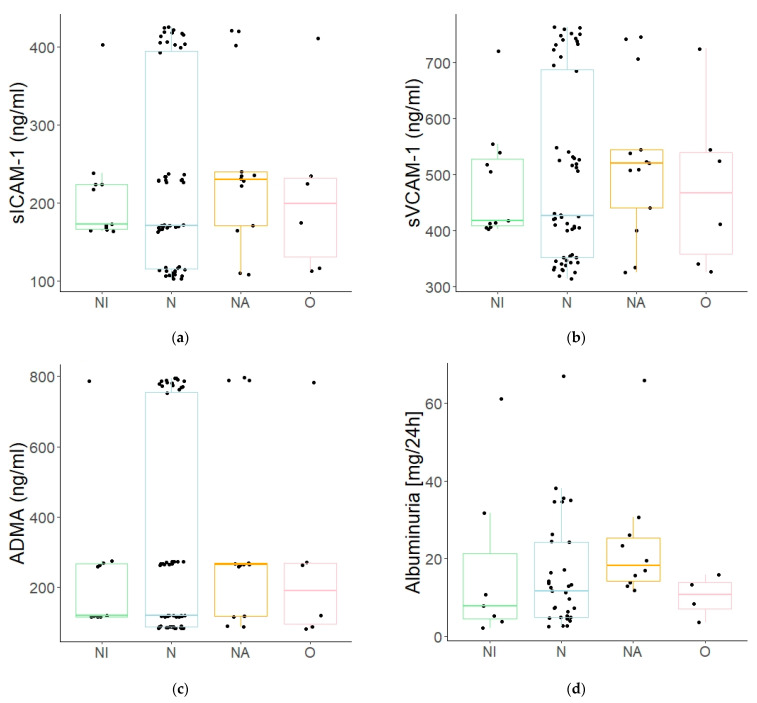
(**a**–**d**). A comparison of selected parameters depending on BMI ((**a**)—the mean sICAM-1 values: 209.94 ± 70.31 vs. 279.44 ± 108.54 vs. 269.65 ± 96.13 vs. 261.02 ± 103.28 ng/mL; (**b**)—the mean sVCAM-1 values: 481.18 ± 99.39 vs. 568.94 ± 141.01 vs. 560.93 ± 117.07 vs. 550.41 ± 129.89 ng/mL; (**c**)—the mean RAGE values: 231.46 ± 198.32 vs. 414.20 ± 298.22 vs. 380.62 ± 269.90 vs. 357.74 ± 291.02 pg/mL; (**d**)—the mean albuminuria values: 17.51 ± 21.68 vs. 15.81 ± 14.41 vs. 23.67 ± 16.08 vs. 10.23 ± 5.41 mg/24 h). Boxplot of values for each subgroup. Legends: NI—the patients underweight patients, *n* = 11; N—the patients with normal body weight, *n* = 39; NA—the overweight patients, *n* = 11; O—the patients with obesity, *n* = 4.

**Figure 4 jcm-10-05065-f004:**
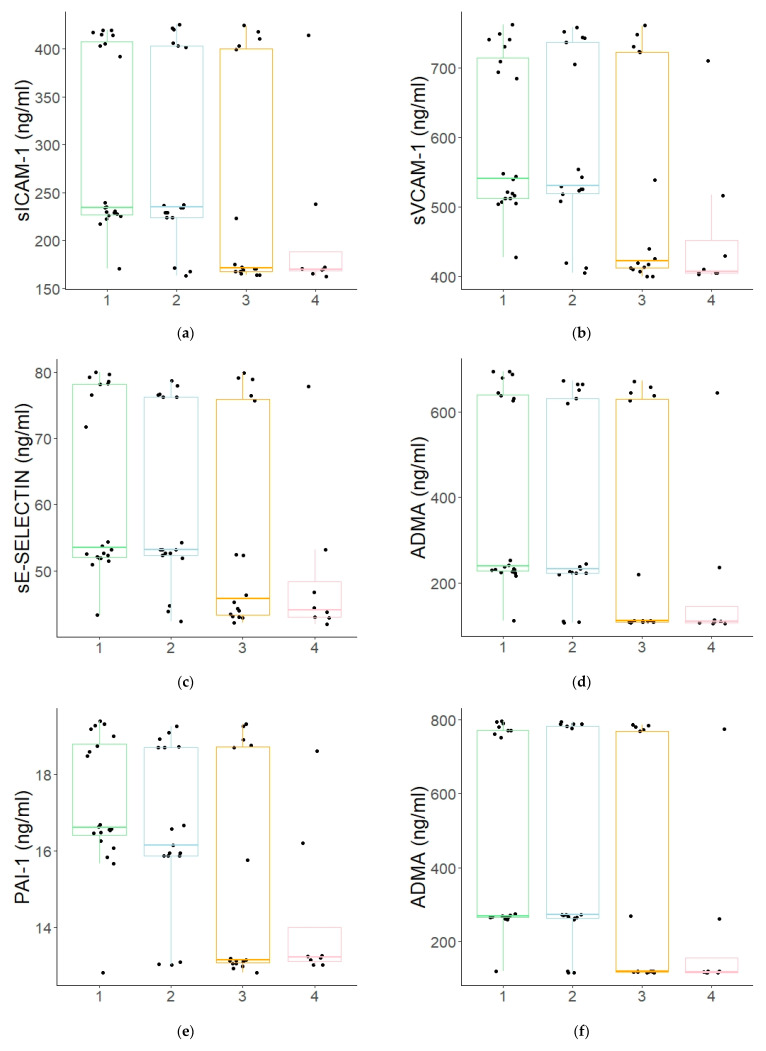
(**a**–**f**). A comparison of endothelial damage markers depending on the degree of metabolic (non)balancing of type 1 diabetes ((**a**)—the mean sICAM-1 values: 298.69 ± 95.06 vs. 284.13 ± 101.18 vs. 247.87 ± 114.65 vs. 207.70 ± 86.90 ng/mL; (**b**)—the mean sVCAM-1 values: 598.80 ± 111.06 vs. 583.04 ± 127.85 vs. 523.53 ± 152.63 vs. 461.07 ± 108.22 ng/mL; (**c**)—the mean sE-SELECTIN values: 62.13 ± 13.35 vs. 59.81 ± 13.58 vs. 55.57 ± 15.92 vs. 49.24 ± 12.07 ng/mL; (**d**)—the mean ADMA values: 397.44 ± 223.47 vs. 356.19 ± 228.50 vs. 284.21 ± 254.30 vs. 191.14 ± 188.38 ng/mL; (**e**)—the mean PAI-1 values: 17.23 ± 1.70 vs. 16.56 ± 2.12 vs. 15.08 ± 2.80 vs. 4.22 ± 2.07 ng/mL; (**f**)—the mean RAGE values: 462.95 ± 265.67 vs. 424.01 ± 281.86 vs. 333.59 ± 312.73 vs. 216.76 ± 231.39 pg/mL). Boxplot of values for each subgroup. Legends: 1—the patients with HbA1c: 6.5–8.9%, *n* = 20; 2—the patients with HbA1c: 9–11.4%, *n* = 17; 3—the patients with HbA1c: 11.5–13.9 %, *n* = 16; 4—the patients with HbA1c: ≥14%, *n* = 8.

**Figure 5 jcm-10-05065-f005:**
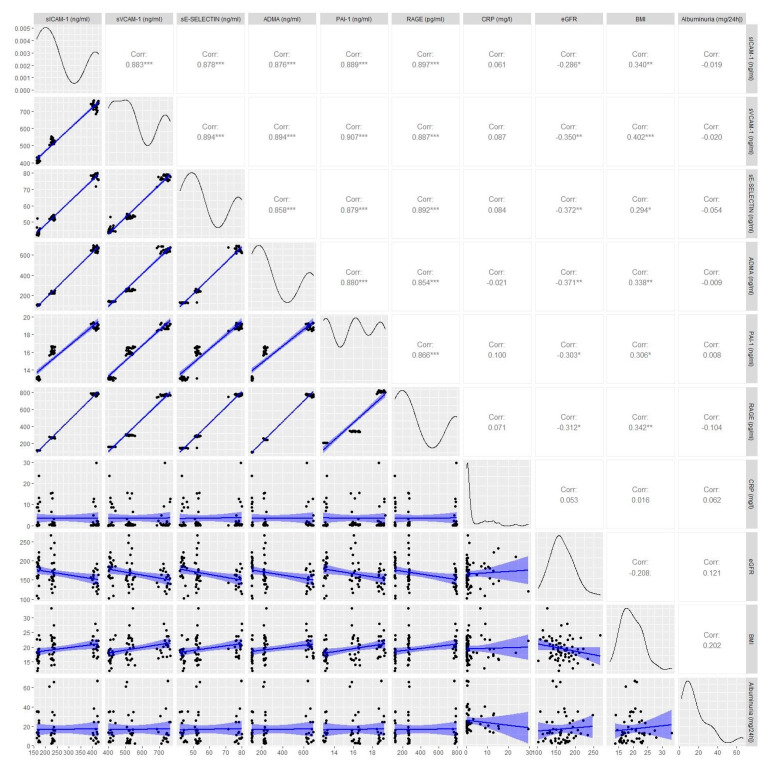
The indices of correlation among selected diagnostic features. Correlogram shows scatterplots for each pair variables in the bottom left part, and Spearman’s correlation coefficient is shown in the upper right part. Distribution of variables is shown diagonally. Legend: *—*p*-value < 0.05; **—*p*-value < 0.01, ***—*p*-value < 0.001.

**Table 1 jcm-10-05065-t001:** Baseline data in diabetic patients (studied group) and in control group.

Parameter	Studied Group *n* = 66		Control Group *n* = 21		*p* Value
	Mean ± SD	Median Value	Mean ± SD	Median Value	
Sex (female, *n* (%))	31 (46.97%)		16 (76.19%)		0.0367
Age (years)	12.69 ± 3.6	13.5	9.26 ± 2.9	9	<0.001
BMI (kg/m^2^)	19.45 ± 3.9	19.02	17.47 ± 2.7	17	0.024
Duration of diabetes (years)	3.8 ± 4.2	2			
HbA1c (%)	10.47 ± 3.07	9			
Haemoglobin (g/dL)	13.21 ± 0.89	13.6	13.21 ± 0.89	13.21	1
Leukocytes (thousand/μL)	6.74 ± 1.81	6.9	6.83 ± 1.71	6.83	0.837
PLT [thousand/μL]	260.66 ± 76.07	261	305.2 ± 75.74	286	0.096
Sodium (mmol/L)	138 ± 3.15	138	138.5 ± 1.36	139	0.402
Potassium (mmol/L)	4.18 ± 0.45	4.2	4.19 ± 0.24	4.19	0.262
CRP [mg/L]	3.54 ± 6.0	0.67	0.77 ± 1.2	0.3	0.011
AspAT	26.33 ± 13.24	24	27.93 ± 5.80	28	0.566
AlAT	17.38 ± 8.53	16	15.73 ± 4.27	14.5	0.273
TSH	2.43 ± 1.45	2.06	2.22 ± 0.85	2.06	0.154
Cholesterol	165.52 ± 32.49	161	172.67 ± 11.67	175	0.104
Triglycerides	98.30 ± 41.53	92	83.5 ± 21.99	83.5	0.089
Urea (mg/dL)	21.13 ± 9.40	22.5	23.55 ± 5.71	23.55	0.353
Creatinine (mg/dL)	0.58 ± 0.14	0.56	0.66 ± 0.09	0.65	0.013
eGFR (mL/min/1.73 m^2^)	166.04 ± 32.86	160.26	114.11 ± 11.00	114.81	<0.001
sICAM-1 (ng/mL)	267.15 ± 111.89	229.2	110.48 ± 4.53	110	<0.001
sVCAM-1 (ng/mL)	554.01 ± 132.11	522.48	337.33 ± 11.73	337.45	<0.001
sE-SELECTIN (ng/mL)	57.74 ± 14.19	52.49	32.64 ± 1.27	32.56	<0.001
ADMA (ng/mL)	325.03 ± 233.39	227.75	71.91 ± 2.31	71.01	<0.001
PAI-1 (ng/mL)	16.03 ± 2.41	16.12	6.51 ± 0.21	6.53	<0.001
RAGE (pg/mL)	380.20 ± 282.90	265.52	84.94 ± 2.27	85.48	<0.001

Legends: SD—standard deviation.

**Table 2 jcm-10-05065-t002:** Demographic, clinical, and biochemical data of the groups depending on DM1 duration Legends: Control—control group; Group 1—the patients with newly diagnosed diabetes mellitus (DM1 duration < 1 year), *n* = 19; Group 2—the patients with DM1 duration of 1–5 years, *n* = 24; Group 3—the patients with DM1 duration of at least 5 years, *n* = 23. ^a^: difference between group I and II is significant (*p* < 0.05), ^b^: difference between Group II and III is significant (*p* < 0.05), ^c^: difference between Group I and III is significant (*p* < 0.05), ^d^: difference between Control group and Group I is significant (*p* < 0.05), ^e^: difference between Control group and Group II is significant (*p* < 0.05), ^f^: difference between Control group and Group III is significant (*p* < 0.05).

Variables.	Control	Group 1	Group 2	Group 3
Age (years)	(9.3 ± 2.9) ^e,f^	(11.2 ± 3.4) ^c^	(11.9 ± 3.8) ^b,e^	(14.8 ± 2.5) ^b,c,f^
BMI (kg/m^2^)	(17.5 ± 2.7) ^f^	(17.3 ± 3.0) ^c^	19.8 ± 4.5	(20.9 ± 3.1) ^c,f^
HbA1c (%)		(12.2 ± 3.2) ^a^	(9.2 ± 2.5) ^a^	10.3 ± 2.9
eGFR mL/min/1.73 m^2^	(114.1 ± 1) ^d,e,f^	(175.5 ± 28.0) ^c,d^	(165.6 ± 33.6) ^e^	(158.4 ± 35.2) ^c,f^
S-creatinine (mg/dL)	(0.6 ± 0.1) ^d,e^	(0.5 ± 0.1) ^c,d^	(0.6 ± 0.1) ^b,e^	(0.6 ± 0.1) ^b,c^
Albuminuria (mg/24 h)		15.2 ± 10.8	14.4 ± 14.2	21.5 ± 19.5
sICAM-1 (ng/mL)	(110.5 ± 4.5) ^d,e,f^	(168.5 ± 3.0) ^a,c,d^	(244.0 ± 55.2) ^a,b,e^	(372.8 ± 85.3) ^b,c,f^
sVCAM-1 (ng/mL)	(337.3 ± 11.7) ^d,e,f^	(412.6 ± 10.3) ^a,c,d^	(538.3 ± 59.8) ^a,b,e^	(687.3 ± 107.8) ^b,c,f^
sE-SELECTIN (ng/mL)	(5.4 ± 0.2) ^d,e,f^	(44.2 ± 2.3) ^a,c,d^	(54.0 ± 6.5) ^a,b,e^	(72.8 ± 11.6) ^b,c,f^
ADMA (ng/mL)	(71.9 ± 2.3) ^d,e,f^	(109.3 ± 2.4) ^a,c,d^	(261.1 ± 124.2) ^a,b,e^	(570.0 ± 193.4) ^b,c,f^
PAI-1 (ng/mL)	(6.5 ± 0.2) ^d,e,f^	(13.1 ± 0.1) ^a,c,d^	(16.3 ± 1.0) ^a,b,e^	(18.2 ± 1.8) ^b,c,f^
RAGE (pg/mL)	(84.9 ± 2.3) ^d,e,f^	(116.8 ± 1.5) ^a,c,d^	(301.9 ± 148.5) ^a,b,e^	(679.5 ± 230.6) ^b,c,f^
CRP (mg/L)	(0.8 ± 1.2) ^e,f^	3.0 ± 4.5	(2.8 ± 4.8) ^e^	(4.8 ± 7.9) ^f^

## Data Availability

The datasets generated and/or analyzed during the current study available from the corresponding author on reasonable request.

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
