# Peer review of "Evaluation of Vascular Endothelial Function in Children with Type 1 Diabetes Mellitus"

_jcm, 2021, doi:10.3390/jcm10215065_

Round 1

Reviewer 1 Report

Here, Nocuń-Wasilewska et al. presented the results of the assessment of selected circulating biomarkers of endothelial dysfunction in a cohort of 66 pediatric patients with type 1 diabetes and matched controls. They provide interesting results on a topic deserving extensive investigation.

A few points need to be addressed:

  • In the introduction, please provide reference that albuminuria is the marker of generalized vascular endothelium damage (line 53)
  • How was type 1 diabetes diagnosed? Were kids tested for fasting insulin levels and, possibly, circulating autoantibodies? If so, is there any relationship between the tested biomarkers and circulating insulin levels?
  • In the Methods section, the paragraph in lines 106-114 is repeated twice (lines 127-135). Please remove the duplicate.
  • The authors should evaluate the levels of the various biomarkers according to the presence (and extent) of albuminuria (e.g. >30 mg/g vs absent/<30 mg/g). Also, in Table 2, they should add albuminuria to the correlation matrix.
  • To this regard, the authors should consider showing ROC curves to assess the diagnostic ability of single or combined biomarkers in type 1 diabetes or diabetic kidney disease. This point would enhance the significance of the paper.
  • Figure quality should be strongly improved by removing unnecessary colors and shadows and by updating the label "Kontrola". Also, please consider removing the title on top of each graph and adding the name of the tested variable on the y-axis. Given the relatively small size of the study cohort, the authors should consider showing individual data points instead of bar charts.

Author Response

Dear Reviewer 1,

Thank you for your apt remarks giving us the opportunity to improve our manuscript.

A few points need to be addressed:

  • In the introduction, please provide reference that albuminuria is the marker of generalized vascular endothelium damage (line 53)

This is corrected by providing reference [Deckert, T., Feldt-Rasmussen, B., Borch-Johnsen, K., Jensen, T., & Kofoed-Enevoldsen, A. (1989). Albuminuria reflects widespread vascular damage. Diabetologia, 32(4), 219-226].

How was type 1 diabetes diagnosed? Were kids tested for fasting insulin levels and, possibly, circulating autoantibodies? If so, is there any relationship between the tested biomarkers and circulating insulin levels?

In all children included into our study, diabetes was diagnosed based on the presence of glycemia ≥ 200 mg / dL [11.1 mmol / L] and the presence of typical symptoms of hyperglycemia (e.g. polydipsia, polyuria, weight loss). Moreover, diagnostics of the type of diabetes in children in our center include tests recommended by ISPAD guidelines (see Pediatric Diabetes October 2018; 19 (Suppl. 27): 7–19), with the assessment of diabetes-associated autoantibodies: glutamic acid decarboxylase 65 autoantibodies (GAD), tyrosine phosphatase-like insulinoma antigen 2 (IA2), insulin autoantibodies (IAA) and β-cell specific zinc transporter 8autoantibodies (ZnT8). Furthermore, levels of fasting blood glucose and fasting c-peptide (as a marker of insulin secretion), as well as HbA1c, are all measured. Each participant underwent the abovementioned tests, the results of which confirmed the initial clinical diagnosis of type 1 diabetes. At the diabetes presentation, all patients had fasting or random hyperglycemia (≥126 mg/dl or ≥200 mg/dl, respectively), all had HbA1c ≥6,5%, all had c-peptide level below the normal range and all were positive for diabetes-associated autoantibodies.
We believe that the assessment of insulinemia itself, in the situation of using insulin in the current treatment of the patient, is redundant, and therefore it is not performed in our center.
The relationship of the tested parameters with the concentration of
c-peptide at presentation nor with the titer of autoantibodies was not assessed in the presented study.

  • In the Methods section, the paragraph in lines 106-114 is repeated twice (lines 127-135). Please remove the duplicate.

The duplicate has been removed.

  • The authors should evaluate the levels of the various biomarkers according to the presence (and extent) of albuminuria (e.g. >30 mg/g vs absent/<30 mg/g). Also, in Table 2, they should add albuminuria to the correlation matrix.

As urine albumin concentration was <30 mg / dl in all studied children, proposed analysis is not feasible. Similarly, it is not possible to derive the ROC curves.

Following paragraph was added

Urine albumin concentration measurements were available for 55 DM1 patients. Median concentration of urine Albumin was 12.9 mg/24h with interquartile range 5.25 mg/24h – 23.85mg/24h. In 10 patients (18.18%) urine albumin concentration exceeded 30 mg/24h. There weren’t any statistically significant differences in levels of biomarkers between groups with and without albuminuria.

Concentration of urine albumin was added to correlation analysis.

  • To this regard, the authors should consider showing ROC curves to assess the diagnostic ability of single or combined biomarkers in type 1 diabetes or diabetic kidney disease. This point would enhance the significance of the paper.

Values of each of the biomarkers don’t overlap between groups with and without type 1 diabetes, thus such ROC curve is unnecessary (minimal value in group with DM1 is higher than maximal value in control group, thus AUC = 1, sensitivity and specificity is 100% for all biomarkers). Instead, a paragraph highlighting possible diagnostic ability was added.

Group of children with diabetic kidney disease was not specifically analyzed in this study. We focused more on eGFR, and provided a linear model of relation between eGFR and biomarkers.

As urine albumin concentration was <30 mg / dl in all studied children it was not possible to derive the ROC curves.

  • Figure quality should be strongly improved by removing unnecessary colors and shadows and by updating the label "Kontrola". Also, please consider removing the title on top of each graph and adding the name of the tested variable on the y-axis. Given the relatively small size of the study cohort, the authors should consider showing individual data points instead of bar charts.

We changed all figures according to your remarks.

Reviewer 2 Report

The study represents a cross-sectional study on endothelial dysfunction markers in children with diabetes mellitus type 1 and its associations with other laboratory markers. The study adds some new novelties to the previous findings; however, better statistical approach is needed to improve it. Overall, my comments on the manuscript are as follows:

Major comments:

  • The type of study should be mentioned in the method section.
  • Does the control group also subjected to surgery as they were hospitalized due to urinary tract abnormalities? If so, did the authors obtain blood prior to surgery? Because surgery can affect the levels of the markers that were checked. Please clarify this issue.
  • Were all the parameters normally distributed which you used Pearson correlation? Please clarify this and if not you must use Spearman test for the data that were not distributed normally.
  • The eGFR was calculated using the old Schwartz formula. Please measure based on the 2009 Schwartz formula which is more accurate (PMID: 19158356). Even you used Jaffe method or creatinine measurements you can use this new equation.
  • The authors should use multiple linear regression to determine the predictive parameters of eGFR among measured biomarkers (sE-selection, ICAM, etc). I believe that simply conclude that sE-selection predicts the eGFR is not correct. You should use modeling with linear regression model and adjust for age, gender, etc and determine the most predictive endothelial dysfunction markers of eGFR.
  • Please add scatter plots of correlation analysis you showed in Table 2 with regression lines.
  • The correlation analysis must be done only on diabetic group. If you did that please clarify in the text. Performing correlation analysis on all subjects (DM and control) is not accurate.

Minor comments:

  • In discussion parts, the following studies which evaluates the endothelial dysfunction markers in children with chronic kidney disease can be discussed and compared with the current results: PMID: 29891747, 20821177, 27563628.
  • The results of figures 1 to 4 can be presented as tables as well, so, it can be more useful.

Author Response

Dear Reviewer 2,

Thank you for your apt remarks giving us the opportunity to improve our manuscript.

Specific comments.

- the Introduction should more efficiently guide the reader to the aims of the study including citing the literature (for example Astrup et al., Diabetes Care, 2008 is important to the subject but missing). Citations are also missing for the biomarkers evaluated in the study and the knowledge gap the study aims to close remains thus only faintly identified.

The introduction was enriched with citations relating to a lack of knowledge whether biomarkers of inflammation and endothelial dysfunction are associated with prognosis in type 1 diabetes

[Astrup, A. S., Tarnow, L., Pietraszek, L., Schalkwijk, C. G., Stehouwer, C. D., Parving, H. H., & Rossing, P. (2008). Markers of endothelial dysfunction and inflammation in type 1 diabetic patients with or without diabetic nephropathy followed for 10 years: association with mortality and decline of glomerular filtration rate. Diabetes Care, 31(6), 1170-1176]

and selected markers of endothelial lesions

[Polek, A., Sobiczewski, W., & Matowicka-Karna, J. (2009). P-selektyna i jej rola w niektórych chorobach P-selectin and its role in some diseases. Postepy Hig Med Dosw.(online), 63, 465-470]

]Chase S. D., Magnani J. L., Simon S. I.: E-selectin ligands as mechanosensitive receptors on neutrophils in health and disease. Annals of Biomedical Engineering 2012; 40(4), 849-859]

[Abbassi O. M. I. D., Kishimoto T. K., McIntire L. V., Anderson D. C., Smith, C. W.: E-selectin supports neutrophil rolling in vitro under conditions of flow. Journal of Clinical Investigation 1993; 92(6), 2719].

- Is it possible that the underlying diseases of the control group (for example urinary tract abnormalities) influenced the urine analysis results?

As it was marked, our control group included children suspected of urinary tract abnormalities or suffering from bedwetting. On the basis of a diagnostic examination, the above-mentioned abnormalities were excluded, because all test results, including markers of inflammation, were negative.

- The abstract appears to have been in a structured format before including it into the unstructured form of the template. The sentences should be formulated to guide through the text.

Corrections have been carried out.

- in general, the manuscript requires language editing.

Corrections have been carried out.

- the introduction could be shortened to the essentials necessary for the scope of the manuscript.

Corrections have been carried out.

- the introduction should also include the current state of the art on endothelial dysfunction and endothelial damage markers in type I diabetic patients and specifically identify the lack of knowledge that led to the present study. Studies close to the topic appear to be missing, 

The study focuses, which is mentioned in  on the need of an identification of reliable markers of progressive endothelium dysfunction to monitor the course of the disease and to diagnose as early as possible any vascular changes in order to prevent their development into serious complications. The more that there is a lack of knowledge whether biomarkers of inflammation and endothelial dysfunction are associated with prognosis in type 1 diabetes.

- the methods section should not report results (age of the study participants etc.)

This is corrected.

- the methodological explanation of the ELISA assays for the endothelial markers as well as the eGFR calculation are redundant in the text (last paragraph page 3 and third paragraph page 4)

This is corrected.

- instead of "SN" (non significant), P-values should be provided

This is corrected whenever possible

- The results section should begin with describing and comparing the study collective. Display of the anthropometric measures and, importantly, characteristics of diabetes (parameters of glucose control, duration of the disease), are missing in the tables, this should all be included.

The missing data is added.

- The authors state that the CRP levels were within normal range (page 5, line 168). However, CRP levels are reported as elevated and significantly different in both study vs. control group, this requires explanation. 

We analyzed changes in the concentration of the studied markers depending on the level of C-reactive protein. It turned out that it correlated with the degree of severity of endothelial inflammation. However, at an early stage of the disease, no significant changes in the concentration of this protein were observed in relation to the control group, while statistically significant differences in the levels of marked markers of endothelial dysfunction were already visible. 

Thus, the phrase: “the levels of C-reactive protein (CRP), the marker of inflammation in the body remained within normal range” was changed into: “the levels of C-reactive protein (CRP), the marker of inflammation in the body, did not differ statistically significantly in relation to the control group (0.8±1.2 vs 2.97±4.53mg/l; p=0.161)

- The graphs require editing (translation into English and further editing, for example, X-axis labeling is missing)

This is corrected

- figure legends should report information on statistics and data displayed (mean and SD?)

This is corrected

- the authors should report the characteristics of the study participant collectives for the subgroups endothelial markers were differentially analyzed in (referring to their DM duration) and not only DM vs. control. This would be important.

Data of the groups depending on DM1 duration (Group I , II, III) and control group (Control) was listed on Table 3

- the discussion begins with a mere explanation of the characteristics of the damage markers that should - drastically shortened and complemented with literature, be move to the introduction.

This is corrected.

- it is of particular significance that the authors find that HbA1c is inversely correlated to the endothelial damage markers. However, the discussion of this finding is not sufficiently discussed: Why would this inverse correlation be expected? If the authors believe this finding can be explained by GFR, the authors need to report GFR values for the subgroups of HBa1c values. How do the authors explain that eFGR linearly correlated with values of endothelial damage and thus prognosis of kidney function?

It should be underlined that not only hyperglycemia has a damaging effect on the endothelium, but also hypoglycemia induces inflammation. [Marfella R, Esposito K, Giunta R, et al. Circulating adhesion molecules in humans: role of hyperglycemia and hyperinsulinemia. Circulation. 2000;101(19):2247-2251. doi:10.1161/01.cir.101.19.2247 ; El Amine M, Sohawon S, Lagneau L, Gaham N, Noordally S. Plasma levels of ICAM-1 and circulating endothelial cells are elevated in unstable types 1 and 2 diabetes [published correction appears in Endocr Regul. 2010 Apr;44(2):76. Langneau, L [corrected to Lagneau, L]; Gaham, N [added]]. Endocr Regul. 2010;44(1):17-24. doi:10.4149/endo_2010_01_17 ; Gogitidze Joy N, Hedrington MS, Briscoe VJ, Tate DB, Ertl AC, Davis SN. Effects of acute hypoglycemia on inflammatory and pro-atherothrombotic biomarkers in individuals with type 1 diabetes and healthy individuals [published correction appears in Diabetes Care. 2010 Sep;33(9):2129]. Diabetes Care. 2010;33(7):1529-1535. doi:10.2337/dc09-0354] Therefore, it is considered that in diabetes the increased risk of chronic complications, resulting from endothelial inflammation, is a consequence of all glycemic fluctuations (so called glycemic variability), having the undoubtedly negative impact on the endothelium. [Kilpatrick ES, Rigby AS, Atkin SL. For debate. Glucose variability and diabetes complication risk: we need to know the answer. Diabet Med. 2010;27(8):868-871. doi:10.1111/j.1464-5491.2010.02929.x ; Ceriello A, Ihnat MA. 'Glycaemic variability': a new therapeutic challenge in diabetes and the critical care setting. Diabet Med. 2010;27(8):862-867. doi:10.1111/j.1464-5491.2010.02967.x Lachin JM, Bebu I, Bergenstal RM, et al. Association of Glycemic Variability in Type 1 Diabetes With Progression of Microvascular Outcomes in the Diabetes Control and Complications Trial. Diabetes Care. 2017;40(6):777-783. doi:10.2337/dc16-2426] This may explain the inverse relationship between HbA1c and the concentration of the investigated markers of inflammation. Frequent episodes of hypoglycaemia lower the level of HbA1c, but at the same time negatively affect the endothelial condition. Furthermore, the majority of children with the highest HbA1c level were the newly diagnosed patients. In those patients the levels of studied parameters were the lowest, indicating the inverse correlation of those with the duration of the disease.

And  we build several LM models however combining several biomarkers didn’t enhance their ability to predict eGFR. Thus we decided to present one with highest statistical significance. Model was then adjusted for age, gender and DM1 duration.

Following paragraph was added:

Simple linear regression was performed to predict eGFR based on each one of studied biomarkers. Among them sE-SELECTIN had the highest R2 =0.09, with regression coefficient = -0.75, SE = 0.28, Intercept = 209.28, P = 0.0088. In multiple linear regression analysis sE-SELECTIN was proven to be independent predictor (P = 0.033) of eGFR. Model was adjusted for age, gender and duration of DM1.

- The discussion requires major language editing and content focus.

This is corrected

Reviewer 3 Report

In this study by Karolina et al., the authors aimed to evaluate endothelial damage markers in type I diabetic children. While the study is of potential interest and the data appears in part interesting, the manuscript and data presentation require major revision. For example, the aim of the study in accordance to the literature should be clearly delineated in the introduction, the methods part requires thorough revision and the subgroup analysis of the results are confusing and insufficiently discussed. On the other hand, major findings (for example the inverse correlation of damage markers with the degree of glycemic control) are only insufficiently evaluated although essential for the research question addressed. The data would benefit from being cleaned up to 2-3 major findings that are clearly delineated, analyzed and discussed. In its present form, data presentation and discussion are confusing and without focus. This reviewer thus recommends de novo submission after major revision.

Specific comments.
- the Introduction should more efficiently guide the reader to the aims of the study including citing the literature (for example Astrup et al., Diabetes Care, 2008 is important to the subject but missing). Citations are also missing for the biomarkers evaluated in the study and the knowledge gap the study aims to close remains thus only faintly identified.
- Is it possible that the underlying diseases of the control group (for example urinary tract abnormalities) influenced the urine analysis results?
- The abstract appears to have been in a structured format before including it into the unstructured form of the template. The sentences should be formulated to guide through the text.
- in general, the manuscript requires language editing.
- the introduction could be shortened to the essentials necessary for the scope of the manuscript.
- the introduction should also include the current state of the art on endothelial dysfunction and endothelial damage markers in type I diabetic patients and specifically identify the lack of knowledge that led to the present study. Studies close to the topic appear to be missing, 
- the methods section should not report results (age of the study participants etc.)
- the methodological explanation of the ELISA assays for the endothelial markers as well as the eGFR calculation are redundant in the text (last paragraph page 3 and third paragraph page 4)
- instead of "SN" (non significant), P-values should be provided
- The results section should begin with describing and comparing the study collective. Display of the anthropometric measures and, importantly, characteristics of diabetes (parameters of glucose control, duration of the disease), are missing in the tables, this should all be included.
- The authors state that the CRP levels were within normal range (page 5, line 168). However, CRP levels are reported as elevated and significantly different in both study vs. control group, this requires explanation. 
- The graphs require editing (translation into English and further editing, for example, X-axis labeling is missing)
- figure legends should report information on statistics and data displayed (mean and SD?)
- the authors should report the characteristics of the study participant collectives for the subgroups endothelial markers were differentially analyzed in (referring to their DM duration) and not only DM vs. control. This would be important.
- the discussion begins with a mere explanation of the characteristics of the damage markers that should - drastically shortened and complemented with literature, be move to the introduction.
- it is of particular significance that the authors find that HbA1c is inversely correlated to the endothelial damage markers. However, the discussion of this finding is not sufficiently discussed: Why would this inverse correlation be expected? If the authors believe this finding can be explained by GFR, the authors need to report GFR values for the subgroups of HBa1c values. How do the authors explain that eFGR linearly correlated with values of endothelial damage and thus prognosis of kidney function?
- The discussion requires major language editing and content focus.

Author Response

Dear Reviewer 3,

Thank you for your apt remarks giving us the opportunity to improve our manuscript.

The study represents a cross-sectional study on endothelial dysfunction markers in children with diabetes mellitus type 1 and its associations with other laboratory markers. The study adds some new novelties to the previous findings; however, better statistical approach is needed to improve it. Overall, my comments on the manuscript are as follows:

Major comments:

  • The type of study should be mentioned in the method section.

The beginning  of Material and Methods is changed in too : “ The prospective observational study involved ….”

  • Does the control group also subjected to surgery as they were hospitalized due to urinary tract abnormalities? If so, did the authors obtain blood prior to surgery? Because surgery can affect the levels of the markers that were checked. Please clarify this issue.

       As it was marked, our control group included children suspected of urinary tract abnormalities or       suffering from bedwetting. On the basis of a diagnostic examination, the above-mentioned abnormalities were excluded, because all test results, including markers of inflammation, were negative.

  • Were all the parameters normally distributed which you used Pearson correlation? Please clarify this and if not you must use Spearman test for the data that were not distributed normally.

Not all parameters were normally distributed, Sperman correlation was used instead.

  • The eGFR was calculated using the old Schwartz formula. Please measure based on the 2009 Schwartz formula which is more accurate (PMID: 19158356). Even you used Jaffe method or creatinine measurements you can use this new equation.

Serum creatinine was determined by the Jaffe method, hence the selection of the old Schwartz formula.

  • The authors should use multiple linear regression to determine the predictive parameters of eGFR among measured biomarkers (sE-selection, ICAM, etc). I believe that simply conclude that sE-selection predicts the eGFR is not correct. You should use modeling with linear regression model and adjust for age, gender, etc and determine the most predictive endothelial dysfunction markers of eGFR.

We build several LM models however combining several biomarkers didn’t enhance their ability to predict eGFR. Thus we decided to present one with highest statistical significance. Model was then adjusted for age, gender and DM1 duration.

Following paragraph was added:

Simple linear regression was performed to predict eGFR based on each one of studied biomarkers. Among them sE-SELECTIN had the highest R2 =0.09, with regression coefficient = -0.75, SE = 0.28, Intercept = 209.28, P = 0.0088. In multiple linear regression analysis sE-SELECTIN was proven to be independent predictor (P = 0.033) of eGFR. Model was adjusted for age, gender and duration of DM1.

  • Please add scatter plots of correlation analysis you showed in Table 2 with regression lines.

Those scatterplots are added in new Table 2.

  • The correlation analysis must be done only on diabetic group. If you did that please clarify in the text. Performing correlation analysis on all subjects (DM and control) is not accurate.

Correlation analysis is performed on DM 1 group.

Minor comments:

  • In discussion parts, the following studies which evaluates the endothelial dysfunction markers in children with chronic kidney disease can be discussed and compared with the current results: PMID: 29891747, 20821177, 27563628.

The following paragraph was added to the discussion:

Musial and Zwolinska reported that the concentrations of matrix metalloproteinases (MMPs) and their tissue inhibitors (TIMPs) correlate not only with the markers of inflammation e.g. e selectin, but also with eGFR, thus indicating increased inflammation and endothelial dysfunction in patients with renal failure. [Musiał K, Zwolińska D. Matrix metalloproteinases (MMP-2,9) and their tissue inhibitors (TIMP-1,2) as novel markers of stress response and atherogenesis in children with chronic kidney disease (CKD) on conservative treatment. Cell Stress Chaperones. 2011;16(1):97-103. doi:10.1007/s12192-010-0214-x] Similar findings were reported by Gheissari et al. and  Meamar et al. [Gheissari A, Meamar R, Abedini A, et al. Association of Matrix Metalloproteinase-2 and Matrix Metalloproteinase-9 With Endothelial Dysfunction, Cardiovascular Disease Risk Factors and thrombotic events in Children With End-stage Renal Disease. Iran J Kidney Dis. 2018;12(3):169-177. ; Meamar R, Shafiei M, Abedini A, et al. Association of E-selectin with hematological, hormonal levels and plasma proteins in children with end stage renal disease. Adv Biomed Res. 2016;5:118. Published 2016 Jul 29. doi:10.4103/2277-9175.186992] As so, it may be assumed that markers of inflammation, i.e. e-selectin, may act not only as the predictors of cardiovascular complications in CKD, but also to predict late diabetes complications, including nephropathy.

  • The results of figures 1 to 4 can be presented as tables as well, so, it can be more useful.

We decided to leave figures but we changed it into a more appropriate form. Important values (that would be included into tables) are presented in relevant paragraphs.

Round 2

Reviewer 2 Report

The authors could successfully address the major concerns in the manuscripts.

Thanks,

Author Response

Dear Reviewer 2,

Comments and Suggestions for Authors

The authors could successfully address the major concerns in the manuscripts.

Thanks,

Thank you again for valuable comments.

Reviewer 3 Report

In this reviewers opinion, the authors have missed to conduct a thorough review of their manuscript. Most and major points raised by the reviewers remained unaddressed. The current version of the manuscript is not acceptable for publication. 

- The introduction remained largely unchanged by the authors, new information is added but not woven in into the existing text. It would be beneficial for the introduction to integrate the new information and provide a quick state-of-the-art overview of what is known on biomarkers of endothelial damage for kidney injury as suggested. Also, the introduction was not shortened.
- the abstract was not changed. It appears this is from a structured version that is not appropriate for J Clin Med format. Sentence 3 is not a whole sentence, the current version has to undergo revision prior to acceptance.
- P-values were not provided.
- the results section does not begin with describing and comparing the study and the control collective as suggested.
- Figures were not edited and changed to English language (for example, "Kontrola").
- Figures are unreadable and not reffered to (what are the scatter plots after Fig. 1?

Author Response

Dear Reviewer 3,

Thank you again for valuable comments.

In this reviewers opinion, the authors have missed to conduct a thorough review of their manuscript. Most and major points raised by the reviewers remained unaddressed. The current version of the manuscript is not acceptable for publication. 

- The introduction remained largely unchanged by the authors, new information is added but not woven in into the existing text. It would be beneficial for the introduction to integrate the new information and provide a quick state-of-the-art overview of what is known on biomarkers of endothelial damage for kidney injury as suggested. Also, the introduction was not shortened.

The introduction was shortened and re-edited in order to incorporate the newly added information into the text.

- the abstract was not changed. It appears this is from a structured version that is not appropriate for J Clin Med format. Sentence 3 is not a whole sentence, the current version has to undergo revision prior to acceptance.

Also the abstract was revised and edited according to the reviewer's suggestions.

- P-values were not provided.

Exact P-values are provided.

- the results section does not begin with describing and comparing the
study and the control collective as suggested.

We changed the results according to your remarks.

- Figures were not edited and changed to English language (for example,
"Kontrola").

Those were figures from an older version of the manuscript. All text in new figures is in English.

- Figures are unreadable and not reffered to (what are the scatter plots
after Fig. 1?

We increased font size in all of the figures and provided adequate description in materials and methods section. All figures and tables were moved to the end of the document for more readability.